# HIV Pre-exposure Prophylaxis (PrEP): Knowledge, attitudes and counseling practices among physicians in Germany – A cross-sectional survey

**Mary Katherine Sammons**[ID]**, Matthew Gaskins**[ID]**, Frank Kutscha, Alexander Nast, Ricardo Niklas Werner**[ID]*

Division of Evidence Based Medicine (dEBM), Department of Dermatology, Venereology and Allergology, Charité –Universitätsmedizin Berlin, Corporate Member of Freie Universität Berlin and Humboldt-Universität zu Berlin, Berlin, Germany

* ricardo.werner@charite.de

## Abstract

### Background

German statutory health insurance began covering the costs associated with HIV PrEP in September 2019; however, to bill for PrEP services, physicians in Germany must either be certified as HIV-specialists according to a nationwide quality assurance agreement, or, if they are non-HIV-specialists, have completed substantial further training in HIV/PrEP care. Given the insufficient implementation of PrEP, the aim of our study was to explore the potential to increase the number of non-HIV-specialists providing PrEP-related services.

### Methods

We conducted an anonymous survey among a random sample of internists, general practitioners, dermatologists and urologists throughout Germany using a self-developed questionnaire. We calculated a knowledge score and an attitudes score from individual items in these two domains. Both scores ranged from 0–20, with high values representing good knowledge or positive attitudes. We also asked participants about the proportion of PrEP advice they provided proactively to men who have sex with men (MSM) and trans-persons who met the criteria to be offered PrEP.

### Results

154 physicians completed the questionnaire. Self-assessed knowledge among HIV-specialists was greater than among non-HIV-specialists [*Median* knowledge score: 20.0 (*IQR* = 0.0) vs. 4.0 (*IQR* = 11.0), *p*<0.001]. Likewise, attitudes towards PrEP were more positive among HIV-specialists than non-HIV-specialists [*Median* attitudes score: 18.0 (*IQR* = 3.0) vs. 13.0 (*IQR* = 5.25), *p*<0.001]. The proportion of proactive advice on PrEP provided to at-risk MSM and trans-persons by HIV-specialists [*Median*: 30.0% (*IQR* = 63.5%)] was higher than that provided by non-HIV-specialists [*Median*: 0.0% (*IQR* = 11.3%), *p*<0.001].

**Data Availability Statement:** All relevant data are within the manuscript and its Supporting Information files: We uploaded the minimal

underlying data set and codebook (Supporting file S3). Age of respondents, postal code, state and qualitative data have been removed to ensure anonymity.

**Funding:** The authors received no specific funding for this work. We acknowledge support from the Open Access Publication Fund of Charité – Universitätsmedizin Berlin.

**Competing interests:** The authors have declared that no competing interests exist.

However, the results of our multiple regression suggest the only independent predictor of proactive PrEP advice was the knowledge score, and not whether physicians were HIV-specialists or non-HIV-specialists.

## Conclusions

These findings point to opportunities to improve PrEP implementation in individuals at risk of acquiring HIV. Targeted training, particularly for non-HIV-specialists, and the provision of patient-centered information material could help improve care, especially in rural areas.

## Introduction

HIV pre-exposure prophylaxis (PrEP) consisting of tenofovir disoproxil fumarate and emtricitabine has been approved for HIV prevention in the United States since 2012 and in the European Union since 2016. Its high effectiveness and safety have been demonstrated in several randomized controlled trials [1–4], and observational studies in a number of metropolitan regions have shown dramatic reductions in the incidence of HIV infections, especially in men who have sex with men (MSM), in recent years–a substantial proportion of which is likely due to PrEP [5–10].

Despite these developments, the uptake of PrEP among those at high risk of HIV acquisition has been slow. By 2019 approximately 224,000 people in the US were estimated to have received a prescription for PrEP, representing only a small fraction of the 1.1 million individuals calculated by researchers at the US Centers for Disease Control and Prevention (CDC) to have an indication for it [11–13]. In Europe, a 2019 study based on data from the European MSM Internet Survey found that an estimated 17.4% of MSM, or 500,000 individuals, in the EU who were very likely to use PrEP were not able to access it [14]. Improving the uptake of PrEP therefore remains a key public health priority.

The German system of statutory health insurance began covering the costs associated with HIV pre-exposure prophylaxis (PrEP) in September 2019. In order to be able to bill for PrEP-related appointments and testing costs, however, physicians in Germany must either be certified according to the German Quality Assurance Agreement on HIV/AIDS as HIV specialists or, if they are non-HIV specialists and belong to certain specialties (e.g., internal and general medicine, dermatology and urology), have completed further training on HIV and PrEP [15]. The training consists of taking part in a 16-hour internship in an outpatient or inpatient HIV care facility and being present during consultations with at least 15 persons who are either living with HIV/AIDS or considering or taking PrEP. In addition, proof of participation in further training courses on the topic must be provided [15]. Because many physicians in Germany work in regions that do not have an outpatient or inpatient HIV treatment facility, the certification requirements represent a substantial barrier to providing PrEP care. This could potentially lead to gaps in treatment, particularly in smaller towns and rural areas, where HIV specialist centers are rare [16].

Given the insufficient implementation of PrEP in populations at risk of acquiring HIV in Germany [17, 18] and beyond, the aim of our study was to explore the potential to increase the number of non-HIV-specialists prescribing PrEP by reducing the barriers to their completing further training. We therefore sought to examine and compare, among HIV-specialists and non-HIV-specialists, self-assessed knowledge and attitudes towards PrEP, as well as the proportion of PrEP advice provided proactively to men who have sex with men (MSM) and trans

persons who met the criteria to be offered PrEP according to the German and Austrian PrEP guideline ("at-risk patients"). Such information could be useful for identifying opportunities to improve PrEP implementation in individuals at risk of acquiring HIV, particularly those in regions underserved by HIV-specialists.

## Materials and methods

### Study design

We conducted a survey among office-based general practitioners, internists, infectious disease specialists, dermato-venereologists and urologists in Germany. Data was collected from August to October 2019. The study was approved by the institutional ethics board of Charité —Universitätsmedizin Berlin (EA1/006/19). Participation was voluntary and no incentives were provided. All participants were older than 18 years. Participants in the online survey provided their written informed consent by ticking the box next to a statement that they had read the study information and agreed to participate in the study. For participants who completed the paper version of the survey, we assumed consent if they returned their questionnaire by fax or mail.

### Setting and participants

Physicians in the abovementioned groups in Germany were eligible to participate in the survey. We classified participants as HIV-specialists if they indicated that they worked in an HIV-specialty practice, and as non-HIV-specialists if they indicated that they did not to work in such a practice. HIV specialist practices in Berlin are owned and staffed primarily by doctors certified as HIV-specialists according to the German Quality Assurance Agreement on HIV/AIDS, and visiting these practices usually requires an appointment. They provide a range of generalist and sexual health care to LGBTI+ people whether or not these individuals are living with HIV.

We used various strategies to recruit participants: (1) We requested the contact details of a random sample of 2,200 office-based physicians in the eligible specialties from the National Association of Statutory Health Insurance Physicians (Kassenärztliche Bundesvereinigung, KBV). We mailed these physicians a paper version of our questionnaire, which could be returned to us by fax or mail. A reminder email with a link to an online version of the questionnaire was sent to the 926 (42%) physicians in this sample for whom we had an email address; (2) An invitation to participate in the survey, containing a link to the online version of the questionnaire, was sent to 253 members of the German AIDS Society (Deutsche AIDS Gesellschaft, DAIG) and to 330 members of the German STI Society (Deutsche STI Gesellschaft, DSTIG) via their online mailing lists. A reminder email was sent two weeks after the initial invitation; (3) Additionally, we placed flyers advertising our study at a Berlin STI conference in September 2019. All online surveys were completely anonymous, with neither IP addresses nor email addresses recorded.

### Variables and measurements

A standardized German-language questionnaire exploring PrEP knowledge, attitudes and counselling practices among physicians in Germany was not available. We therefore developed the questionnaire for the purposes of the present study (S1 and S2 Files). The original draft questionnaire (MS) was tested and discussed (RW, MG, FK) to identify and solve any problems concerning the comprehensibility of the content and design, and to ensure alignment with a related questionnaire we developed to explore PrEP knowledge, attitudes and counselling practices among non-governmental counselling centres and local health offices in Germany. The results of this latter study are published elsewhere [19].

Demographic data included medical specialty, whether the practice had been certified according to the Quality Assurance Agreement on HIV/AIDS, age, gender and languages spoken. The first three numbers of the practice zip codes were recorded to determine in which of Germany's 16 states the practice was located. To obtain contextual information about the practice, we asked how many (a) HIV tests had been performed, (b) HIV infections diagnosed and (c) MSM and transgender patients seen within an average calendar quarter (3 months).

After providing a brief summary of the recommendations of the German and Austrian guideline [20] on the indications for offering PrEP to HIV-negative MSM and transgender persons (which served as our definition of "at-risk patients"), we asked participants to indicate the number of patients they saw during an average quarter who fulfilled these criteria and the number of these patients who were provided with advice on PrEP proactively by the physician. Self-assessed knowledge about PrEP and self-reported attitudes towards PrEP were quantified as described in our previous study [19]. This comprised the calculation of a summative knowledge score and a summative attitudes score from five individual knowledge and attitude items, respectively. The total scores ranged from 0 and 20, with high values representing good knowledge or positive attitudes toward PrEP, respectively. Furthermore, we presented a list of various aspects that might be perceived as barriers to patients initiating PrEP and asked participants to rate the relevance of each of these aspects on an 11-level rating scale. This included barriers for the patients as assessed in the previous study [19], as well as additional barriers for physicians. Lastly, we asked participants which training or information materials would help them with PrEP advice and prescriptions [19].

## Sample size and statistical methods

The questionnaire was developed for the purposes of this study, and no data were available on expected means or variability. Therefore, no sample size calculation was performed and the size of the random sample (n = 2,200) was based on feasibility considerations. Statistical analyses were performed using IBM® SPSS® Statistics version 25 (sample characteristics and bivariate statistics) and STATA SE version 14.2 (linear regression). Independent t-tests, Mann-Whitney U-tests, Pearson's chi squared tests and Fisher's Exact tests were used to quantify associations between variables, depending on the distribution and type of data.

We performed a multiple linear regression using both backward and forward elimination to identify predictors of the proportion of proactive advice on PrEP that had been provided during appointments with at-risk patients. The following variables for the regression model were purposefully selected a priori: HIV specialist status (HIV-specialists vs. non-HIV-specialists), size of the city in which the physician's practice was located, location in either a western or eastern German state (with eastern states being defined as any of the five new states formed from the territory of former East Germany as part of German reunification in 1990), gender, percentage of positive HIV tests (number of positive tests/total number of patients tested), knowledge score and attitudes score. The stopping rule for eliminating individual variables in the logistic regression was $p < 0.2$. Variance inflation factor (VIF) statistics, tolerance and condition index were used to ensure that there was no multi-collinearity of the predictors or instability of the regression coefficients. Missing cases were excluded in a listwise fashion. The level for statistical significance was set at $p < 0.05$.

## Results

### Demographic data

We received a total of 161 responses, of which we excluded seven because they did not provide meaningful information. The sample included in our analyses therefore consisted of 154

respondents, 72 of whom indicated that they worked in an HIV-specialty practice and 79 of whom indicated that they did not work in such a practice ("non-HIV-specialists"). Three participants did not provide information about their HIV specialist status or medical specialty; data from these participants were included only in the analyses of barriers to the prescription of PrEP and of helpful materials and training. Demographic data of the sample, including tests for differences according to HIV specialist status, are shown in *Table 1*. Statistically significant associations between HIV specialist status and demographic data were found for gender ($\chi^2$($df$ = 1, $n$ = 151) = 6.938, $p$ = 0.008), specialty ($\chi^2$($df$ = 5, $n$ = 151) = 83.379, $p$<0.001), size of the city in which the practice was located ($\chi^2$($df$ = 3, $n$ = 142) = 33.378, $p$<0.001), and the state in which the practice was located (i.e., eastern states vs. western states) ($\chi^2$($df$ = 1, $n$ = 142) = 3.833, $p$ = 0.05).

## Physician appointments with at-risk patients and HIV testing practice

*Table 2* depicts data on the number of (a) appointments with MSM and trans persons overall, (b) appointments with MSM and trans persons who met the criteria to be offered PrEP according to the German and Austrian guideline ("at-risk patients"), (c) the overall number of HIV tests and (d) the number and proportion of positive HIV tests per quarter as indicated by the respondents. For all of the mentioned variables, we found statistically significant differences between HIV-specialists and non-HIV-specialists.

Independent of their HIV specialist status, the respondents indicated that in a median of 15.5% of their appointments with at-risk patients, they proactively provided advice on PrEP (*Table 3*). The proportion of appointments with at-risk patients in which the physician provided proactive advice on PrEP was significantly higher among HIV-specialists than it was among non-HIV-specialists: HIV-specialists indicated that they proactively provided advice on PrEP in a median of 30.0% of their contacts with at-risk patients, whereas non-HIV-specialists indicated that they proactively provided advice on PrEP in a median of 0.0% of their contacts, $U$ = 468.500, $p$<0.001.

## Self-assessment of PrEP knowledge and advising competence

For each of the self-assessed dimensions of knowledge and competence, the participants in our survey tended to agree with the relevant statements in the questionnaire if they were HIV-specialists, whereas they tended to disagree with these statements if they were non-HIV-specialists. These differences were found to be statistically significant (*Table 4*). Correspondingly, the summative knowledge score was higher for HIV-specialists (*Median* = 20.0, *IQR* = 0.0) than it was for non-HIV-specialists (*Median* = 4.0, *IQR* = 11.0), $U$ = 279.0, $p$<0.001.

## Attitudes towards PrEP

Regarding attitudes towards PrEP, we found that HIV-specialists agreed with all of the statements expressing a positive attitude and disagreed with the statement expressing a negative attitude more often than the non-HIV-specialists (*Table 5*). As with the summative knowledge score reported above, the summative attitudes score was higher among HIV-specialists (*Median* = 18.0, *IQR* = 3.0) than among non-HIV-specialists (*Median* = 13.0, *IQR* = 5.25), $U$ = 588, $p$<0.001.

## Multiple linear regression on the proportion of proactive PrEP advice

To determine independent factors that predicted the proportion of PrEP advice provided proactively by physicians to at-risk patients, we developed a multiple linear regression model. Applying both a backward elimination and a stepwise forward elimination method (both with a stopping rule of p<0.2 for the exclusion or inclusion of each variable), the same regression

**Table 1. Demographic data and contextual characteristics of the sample.**

| Variable | Total sample | | HIV specialist status | | | |
|---|---|---|---|---|---|---|
| | | | HIV-specialists | | Non-HIV-specialists | |
| N | 154* | | 72 | | 79 | |
| **Age in years** (n = 145) | | | | | | $p = 0.180^\dagger$ |
| Mean (SD) | 52.22 | (8.98) | 51.20 | (8.46) | 53.20 | (9.39) |
| Min; Max | 33–84 | | 34–76 | | 33–84 | |
| **Gender** (n, %) | | | | | | $p = 0.008^\S$ |
| Female | 54 | (35.1%) | 18 | (25.0%) | 36 | (45.6%) |
| Male | 97 | (63.0%) | 54 | (75.0%) | 43 | (54.4%) |
| Not specified | 3 | (1.9%) | 0 | (0.0%) | 0 | (0.0%) |
| **Specialty** (n, %) | | | | | | $p < 0.001^\S$ |
| General Medicine | 35 | (22.7%) | 11 | (15.3%) | 24 | (30.4%) |
| Internal Medicine | 27 | (17.5%) | 22 | (30.6%) | 5 | (6.3%) |
| Dermatology | 25 | (16.2%) | 4 | (5.6%) | 21 | (26.6%) |
| Urology | 25 | (16.2%) | 0 | (0.0%) | 25 | (31.6%) |
| General Medicine and Internal Medicine with Additional Qualification for Infectious Disease | 37 | (24.0%) | 35 | (48.6%) | 2 | (2.5%) |
| Not specified | 5 | (3.4%) | 0 | (0.0%) | 2 | (2.5%) |
| **Size of city** (n, %) | | | | | | $p < 0.001^\S$ |
| Metropolis (>1,000,000) | 52 | (33.8%) | 36 | (50.0%) | 16 | (20.3%) |
| Large city (>100,000) | 44 | (28.6%) | 25 | (34.7%) | 19 | (24.1%) |
| City (>10,000) | 27 | (17.5%) | 4 | (5.6%) | 23 | (29.1%) |
| Small city (≤ 10,000) | 19 | (12.3%) | 2 | (2.8%) | 17 | (21.5%) |
| Not specified | 12 | (7.8%) | 5 | (6.9%) | 4 | (5.1%) |
| **State** (n, %) | | | | | | $p = 0.05^\#$ |
| **Western German states, including Berlin** | **123** | **(79.9%)** | **62** | **(86.1%)** | **61** | **(77.2%)** |
| Baden-Wuerttemberg | 15 | (9.7%) | 8 | (11.1%) | 7 | (8.9%) |
| Bavaria | 18 | (11.7%) | 13 | (18.1%) | 5 | (6.3%) |
| Berlin | 26 | (16.9%) | 14 | (19.4%) | 12 | (15.2%) |
| Bremen | 2 | (1.3%) | 0 | (0%) | 2 | (2.5%) |
| Hamburg | 5 | (3.2%) | 4 | (5.6%) | 1 | (1.3%) |
| Hesse | 23 | (14.9%) | 12 | (16.7%) | 11 | (13.9%) |
| Lower Saxony | 5 | (3.2%) | 0 | (0.0%) | 5 | (6.3%) |
| North Rhine-Westphalia | 22 | (14.3%) | 10 | (13.9%) | 12 | (15.2%) |
| Rhineland-Palatinate | 5 | (3.2%) | 1 | (1.4%) | 4 | (5.1%) |
| Saarland | 2 | (1.3%) | 0 | (0.0%) | 2 | (2.5%) |
| Schleswig-Holstein | 0 | (0.0%) | 0 | (0.0%) | 0 | (0.0%) |
| **Eastern German states, excluding Berlin** | **19** | **(12.3%)** | **5** | **(6.9%)** | **14** | **(17.7%)** |
| Brandenburg | 2 | (1.3%) | 0 | (0.0%) | 2 | (2.5%) |
| Mecklenburg-Western Pomerania | 1 | (0.6%) | 0 | (0.0%) | 1 | (1.3%) |
| Saxony | 7 | (4.5%) | 3 | (4.2%) | 4 | (5.1%) |
| Saxony-Anhalt | 5 | (3.2%) | 0 | (0.0%) | 5 | (6.3%) |
| Thuringia | 4 | (2.6%) | 2 | (2.8%) | 2 | (2.5%) |
| Not specified | 12 | (7.8%) | 5 | (6.9%) | 4 | (5.1%) |

Max, maximum; Min, minimum; SD, standard deviation;

*3 patients who were included in some of the analyses in the present study did not provide information about their specialist status (HIV-specialists vs. non-HIV-specialists);

†From independent samples t-tests of the null hypothesis that the mean value of non-HIV-specialists is equal to that of HIV specialists;

§From Pearson's Chi squared tests of the null hypothesis that there is no statistically significant difference between the observed and expected frequencies in each category, according to the HIV specialist status;

#From Pearson's Chi squared tests of the null hypothesis that there is no statistically significant difference between the observed and expected frequencies in the categories "western German states" vs. "eastern German states", according to the HIV specialist status.

**Table 2. Number of appointments with different categories of patients and HIV-tests per calendar quarter.**

| | | | HIV specialist status | | |
|---|---|---|---|---|---|
| Variable | | Total sample | HIV-specialists | | Non-HIV-specialists |
| **Number of overall appointments with MSM and trans persons per quarter** ($n = 141$) | | | | | $p < 0.001$[†] |
| | *Median* (IQR) | 30.0 (345.0) | 375.0 (400.0) | | 5.0 (18.0) |
| | *Mean* (SD) | 162.50 (213.05) | 327.88 (210.47) | | 16.97 (33.20) |
| | *Q1 –Q3* | 5.0–350.0 | 100.0–500.0 | | 2.0–20.0 |
| **Number of appointments with MSM and trans persons who met the criteria to be offered PrEP according to the German and Austrian guideline (at-risk clients) per quarter** ($n = 131$) | | | | | $p < 0.001$[†] |
| | *Median* (IQR) | 17.0 (99.0) | 100.0 (170.0) | | 1.0 (6.0) |
| | *Mean* (SD) | 71.74 (114.08) | 143.60 (132.33) | | 7.17 (15.33) |
| | *Q1 –Q3* | 1.0–100.0 | 30.0–200.0 | | 0.0–6.0 |
| **Overall number of HIV tests per quarter** ($n = 145$) | | | | | $p < 0.001$[†] |
| | *Median* (IQR) | 20.0 (87.0) | 80.0 (195.0) | | 4.0 (17.7) |
| | *Mean* (SD) | 73.14 (124.03) | 139.94 (152.79) | | 12.50 (23.21) |
| | *Q1 –Q3* | 3.0–90.0 | 30.0–225.0 | | 1.0–18.7 |
| **Number of positive HIV test results per quarter** ($n = 143$) | | | | | $p < 0.001$[†] |
| | *Median* (IQR) | 1.0 (2.0) | 2.0 (4.0) | | 0.0 (1.0) |
| | *Mean* (SD) | 5.64 (30.46) | 11.45 (43.93) | | 0.51 (1.36) |
| | *Q1 –Q3* | 0.0–2.0 | 1.0–5.0 | | 0.0–1.0 |
| **Proportion of positive HIV test results among overall number of HIV tests per quarter** ($n = 140$) | | | | | $p < 0.001$[†] |
| | *Median* (IQR) | 1.63% (6.50%) | 2.83% (8.73%) | | 0.00% (5.00%) |
| | *Mean* (SD) | 6.47% (12.41%) | 8.02% (10.16%) | | 5.16% (13.96%) |
| | *Q1 –Q3* | 0.00%-6.50% | 1.27%-10.00% | | 0.00%-5.00% |

*IQR*, interquartile range; *Q1*, first quartile; *Q3*, third quartile;

[†]From Mann-Whitney U-tests of the null hypothesis that the median value of HIV specialists is equal to that of non HIV specialists.

equation was found ($F(3,79) = 7.70$, $p<0.001$, $n = 83$), with $R^2 = 0.165$ (*Table 6*). Only the city size, knowledge score and attitudes score remained in the model; ultimately, however, the only statistically significant predictor was the knowledge score.

## Educational materials and barriers

In total, 121 participants answered the question about which materials or tools they thought would increase the practicability of their PrEP counselling and prescriptions. Patient decision

**Table 3. Advice on PrEP during appointments with MSM and trans persons who met the criteria to be offered PrEP according to the German and Austrian guideline (at-risk patients).**

| | | Total sample | HIV specialist status | | |
|---|---|---|---|---|---|
| Variable | | | HIV-specialists | | Non-HIV-specialists |
| **Proportion of appointments with 'at-risk' MSM and trans persons in which physicians themselves proactively address the topic PrEP** ($n = 102$) | | | | | $p < 0.001$[†] |
| | *Median* (IQR) | 15.48% (50.0%) | 30.00% (63.50%) | | 0.00% (11.32%) |
| | *Mean* (SD) | 30.20% (35.34%) | 40.70% (34.21%) | | 16.36% (32.21%) |
| | *Q1 –Q3* | 0.00% - 50.00% | 11.50% - 75.00% | | 0.00% - 11.32% |

*IQR*, interquartile range; *Q1*, first quartile; *Q3*, third quartile; *SD*, standard deviation;

[†]From Mann-Whitney U-tests of the null hypothesis that the median value of HIV-specialists is equal to that of non-HIV-specialists.

**Table 4. Self-assessment of knowledge and counselling competence.**

| Variable | Total sample | | HIV specialist status | | | |
|---|---|---|---|---|---|---|
| | | | HIV-specialists | | Non-HIV-specialists | |
| **Global assessment: "I am well-informed about PrEP" (n, %), *n* = 128** | | | | | | *p* < 0.001[†] |
| Strongly disagree | 31 | (24.2%) | 1 | (1.8%) | 30 | (42.3%) |
| Disagree | 17 | (13.3%) | 0 | (0.0%) | 17 | (23.9%) |
| Neither agree nor disagree | 6 | (4.7%) | 1 | (1.8%) | 5 | (7.0%) |
| Agree | 16 | (12.5%) | 4 | (7.0%) | 12 | (16.9%) |
| Strongly agree | 58 | (45.3%) | 51 | (89.5%) | 7 | (9.9%) |
| **Indications: "I am able to comprehensively give patients advice on whether it makes sense to take PrEP in their respective case" (n, %), *n* = 128** | | | | | | *p* < 0.001[†] |
| Strongly disagree | 23 | (18.0%) | 1 | (1.8%) | 22 | (31.0%) |
| Disagree | 22 | (17.2%) | 0 | (0.0%) | 22 | (31.0%) |
| Neither agree nor disagree | 10 | (7.8%) | 1 | (1.8%) | 9 | (12.7%) |
| Agree | 15 | (11.7%) | 5 | (8.8%) | 10 | (14.1%) |
| Strongly agree | 58 | (45.3%) | 50 | (87.7%) | 8 | (11.3%) |
| **Adverse effects: "I am able to comprehensively give patients advice on the adverse effects of PrEP" (n, %), *n* = 128** | | | | | | *p* < 0.001[†] |
| Strongly disagree | 31 | (24.2%) | 1 | (1.8%) | 30 | (42.3%) |
| Disagree | 19 | (14.8%) | 0 | (0.0%) | 19 | (26.8%) |
| Neither agree nor disagree | 7 | (5.5%) | 0 | (0.0%) | 7 | (9.9%) |
| Agree | 11 | (8.6%) | 3 | (5.3%) | 8 | (11.3%) |
| Strongly agree | 60 | (46.9%) | 53 | (93.0%) | 7 | (9.9%) |
| **Modalities of intake: "I am able to comprehensively give patients advice on the possible modalities of intake of PrEP (e.g., continuous vs. on-demand)" (n, %), *n* = 128** | | | | | | *p* < 0.001[†] |
| Strongly disagree | 31 | (24.2%) | 1 | (1.8%) | 30 | (42.3%) |
| Disagree | 20 | (15.6%) | 0 | (0.0%) | 20 | (28.2%) |
| Neither agree nor disagree | 5 | (3.9%) | 1 | (1.8%) | 4 | (5.6%) |
| Agree | 10 | (7.8%) | 2 | (3.5%) | 8 | (11.3%) |
| Strongly agree | 62 | (48.4%) | 53 | (93.0%) | 9 | (12.7%) |
| **Investigations: "I am able to comprehensively give patients advice on the medical investigations necessary during the use of PrEP" (n, %), *n* = 128** | | | | | | *p* < 0.001[†] |
| Strongly disagree | 29 | (22.7%) | 1 | (1.8%) | 28 | (39.4%) |
| Disagree | 20 | (15.6%) | 0 | (0.0%) | 20 | (28.2%) |
| Neither agree nor disagree | 6 | (4.7%) | 1 | (1.8%) | 5 | (7.0%) |
| Agree | 9 | (7.0%) | 2 | (3.5%) | 7 | (9.9%) |
| Strongly agree | 64 | (50.0%) | 53 | (93.0%) | 11 | (15.5%) |
| **Knowledge score (0–20), *n* = 128** | | | | | | *p* < 0.001[#] |
| *Median* (IQR) | 15.0 | (17.0) | 20.0 | (0.0) | 4.0 | (11.0) |
| *Mean* (SD) | 11.89 | (8.43) | 19.23 | (2.96) | 6.49 | (6.76) |
| *Q1 –Q3* | 3.0–20.0 | | 20.0–20.0 | | 0.0–11.0 | |

*Max*, maximum; *Min*, minimum; *IQR*, interquartile range; *Q1*, first quartile; *Q3*, third quartile;

[†]From Fisher's Exact tests of the null hypothesis that there is no statistically significant difference between the observed and expected frequencies in each category, by physician group.

[#]From Mann-Whitney U-tests of the null hypothesis that the median value of HIV-specialists is equal to that of non-HIV-specialists.

aids that present information on PrEP in patient-friendly language (71.9%, *n* = 87) and in different languages (56.2%, *n* = 68) were chosen most frequently. About half of the respondents (53.7%, *n* = 65) indicated that a national guideline containing a clear presentation of indications, contraindications and laboratory investigations would be helpful. Whereas about half of

**Table 5. Attitudes towards PrEP.**

| Variable | Total sample | | HIV specialist status | | | |
|---|---|---|---|---|---|---|
| | | | HIV-specialists | | Non-HIV-specialists | |
| **Global assessment: "I think that PrEP is an important element of HIV prevention strategies" (n, %), _n_ = 126** | | | | | | _p_ < 0.001[§] |
| Strongly disagree | 1 | (0.8%) | 0 | (0.0%) | 1 | (1.4%) |
| Disagree | 7 | (5.6%) | 1 | (1.8%) | 6 | (8.7%) |
| Neither agree nor disagree | 10 | (7.9%) | 1 | (1.8%) | 9 | (13.0%) |
| Agree | 30 | (23.8%) | 4 | (7.0%) | 26 | (37.7%) |
| Strongly agree | 78 | (61.9%) | 51 | (89.5%) | 27 | (39.1%) |
| **Reliability: "I think that PrEP is a reliable method to protect oneself from HIV" (n, %), _n_ = 124** | | | | | | _p_ < 0.001[§] |
| Strongly disagree | 5 | (4.0%) | 0 | (0.0%) | 5 | (7.5%) |
| Disagree | 8 | (6.5%) | 0 | (0.0%) | 8 | (11.9%) |
| Neither agree nor disagree | 19 | (15.3%) | 4 | (7.0%) | 15 | (22.4%) |
| Agree | 44 | (35.5%) | 16 | (28.1%) | 28 | (41.8%) |
| Strongly agree | 48 | (38.7%) | 37 | (64.9%) | 11 | (16.4%) |
| **Adverse effects: "I think that PrEP is a method to protect oneself from HIV that has few side effects" (n, %), _n_ = 124** | | | | | | _p_ < 0.001[§] |
| Strongly disagree | 5 | (4.0%) | 0 | (0.0%) | 5 | (7.4%) |
| Disagree | 19 | (15.3%) | 2 | (3.6%) | 17 | (25.0%) |
| Neither agree nor disagree | 36 | (29.0%) | 11 | (19.6%) | 25 | (36.8%) |
| Agree | 37 | (29.8%) | 21 | (37.5%) | 16 | (23.5%) |
| Strongly agree | 27 | (21.8%) | 22 | (39.3%) | 5 | (7.4%) |
| **Availability of better alternatives: "I think that PrEP is unnecessary, because there are better alternatives to protect oneself from HIV" (n, %), _n_ = 121** | | | | | | _p_ = 0.003[§] |
| Strongly disagree | 54 | (44.6%) | 34 | (59.6%) | 20 | (31.3%) |
| Disagree | 38 | (31.4%) | 17 | (29.8%) | 21 | (32.8%) |
| Neither agree nor disagree | 23 | (19.0%) | 5 | (8.8%) | 18 | (28.1%) |
| Agree | 3 | (2.5%) | 1 | (1.8%) | 2 | (3.1%) |
| Strongly agree | 3 | (2.5%) | 0 | (0.0%) | 3 | (4.7%) |
| **Reimbursement of costs: "I think that PrEP should be paid for by the statutory health insurance" (n, %), _n_ = 124** | | | | | | _p_ = 0.001[§] |
| Strongly disagree | 10 | (8.1%) | 1 | (1.8%) | 9 | (13.4%) |
| Disagree | 15 | (12.1%) | 3 | (5.3%) | 12 | (17.9%) |
| Neither agree nor disagree | 23 | (18.5%) | 10 | (17.5%) | 13 | (19.4%) |
| Agree | 25 | (20.2%) | 9 | (15.8%) | 16 | (23.9%) |
| Strongly agree | 51 | (41.1%) | 34 | (59.6%) | 17 | (25.4%) |
| **Attitude Score (0–20), _n_ = 118** | | | | | | _p_ < 0.001[†] |
| _Median_ (IQR) | 15.5 | (5.0) | 18.0 | (3.0) | 13.0 | (5.25) |
| _Mean_ (SD) | 14.93 | (3.92) | 17.29 | (2.59) | 12.90 | (3.78) |
| _Q1 –Q3_ | 13.0–18.0 | | 16.0–19.0 | | 10.0–15.25 | |

_IQR_, interquartile range; _Q1_, first quartile; _Q3_, third quartile;

[†]From Mann-Whitney U-tests of the null hypothesis that the median value of HIV-specialists is equal to that of non-HIV-specialists

[§]From Fisher's Exact tests of the null hypothesis stating that there is no statistically significant difference between the observed and expected frequencies in each category, according to physician group.

the respondents (53.7%, _n_ = 65) indicated that educational material or training on the management of PrEP would be useful for their practice, fewer indicated that educational material or training on identifying PrEP candidates (38.8%, _n_ = 47) or on talking with patients about sex (29.8%, _n_ = 36) would be helpful. However, significantly more non-HIV-specialists than HIV-specialists indicated that they wished to receive educational material or training on how to

**Table 6. Multiple linear regression to predict the proportion of PrEP advice provided proactively to MSM and trans persons who meet the criteria be offered PrEP according to the German and Austrian guideline (at-risk patients).**

| Predictors | Coefficient (*Robust SE*) | | Beta | *p* | *VIF* |
|---|---|---|---|---|---|
| Constant | -32.632 | (16.238) | | 0.048 | |
| Size of the city [1] | 6,107 | (4.553) | 0.170 | 0.184 | 1.39 |
| Knowledge score[2] | 1,782 | (0.585) | 0.320 | 0.003 | 2.00 |
| Attitudes score[3] | 1,851 | (1.031) | 0.191 | 0.077 | 1.57 |

*SE*, standard error; *VIF*, variance inflation factor; [1] Size of the city coded in 4 categories with 0 indicating more than 1,000,000 inhabitants and 3 indicating less than 10,000 inhabitants [2] Scale from 0 to 20 points, with higher scores indicating a more positive self-assessment of knowledge about PrEP and counselling competence; [3] Scale from 0 to 20 points, with higher scores indicating a more positive attitude towards PrEP.

manage PrEP users (61.9% vs. 43.6%, $\chi^2(df = 1, n = 118) = 3.938, p = 0.047$) and to identify PrEP candidates (50.8% vs. 25.5%, $\chi^2(df = 1, n = 118) = 7.926, p = 0.005$). Less than half of the respondents (45.5%, $n = 55$) indicated that an app- or text-message-based reminder service for patients would be useful to increase the adherence of PrEP users.

When respondents were asked to rate the relevance of barriers for patients to initiate PrEP, they rated the following as the most relevant: patients underestimating their own risk of acquiring HIV infection (*Median* = 8.00, *IQR* = 4.0), difficulties in finding a doctor to pre-scribe PrEP (*Median* = 8.00, *IQR* = 5.5) and the time required for regular visits to the doctor (*Median* = 7.0, *IQR* = 6.0). Further results on perceived barriers to PrEP initiation and their relevance for patients are shown in *Table 7*. Among the barriers for physicians, respondents indicated that time-consuming management of PrEP patients was a relevant barrier (*Median* = 7.0, *IQR* = 4.0), but that difficulties identifying those who would benefit from PrEP were less relevant (*Median* = 3.0, *IQR* = 6.0).

## Discussion

Our study is the first of its kind to assess physicians' knowledge of HIV PrEP, their attitudes towards it, and their counseling practices in consultations with patients across Germany who are interested in or have indications for PrEP. Given the large gap, in the EU and beyond, between individuals who are interested in using PrEP but are unable to access it, we aimed to explore with our survey whether there might be potential to increase the number of non-HIV-

**Table 7. Barriers for patients to initiate PrEP as perceived by participating physicians.**

| | *n* | *Median* (*IQR*) | |
|---|---|---|---|
| Assessment of the own risk of getting infected with HIV as too low to take PrEP | 69 | 8.0 | (4.0) |
| Difficulties finding a doctor who prescribes PrEP | 74 | 8.0 | (5.5) |
| Time required for regular visits to the doctor | 66 | 6.0 | (6.0) |
| The monthly costs of the PrEP medication | 69 | 6.0 | (6.0) |
| Lack of information about PrEP in patient-friendly language | 68 | 5.0 | (5.0) |
| Lack of information about PrEP in the native language of the client | 68 | 5.0 | (5.0) |
| Worries about getting infected with other STIs | 71 | 5.0 | (5.0) |
| Cultural barriers | 72 | 5.0 | (6.0) |
| The costs of the laboratory tests | 73 | 5.0 | (6.0) |
| Worries about severe or permanent side effects | 68 | 4.0 | (5.0) |
| Worries about mild or temporary side effects | 67 | 3.0 | (4.0) |
| Worries about stigmatization in the peer group | 69 | 3.0 | (5.0) |

specialists providing PrEP-related services in Germany by reducing the barriers to their completing further training and thus being able to bill for these services.

It is therefore highly relevant that participants in our survey rated "difficulties in finding a doctor who prescribes PrEP" as one of the most important barriers for patients to initiate chemoprophylaxis. The lack of HIV-specialists in rural areas is well-reflected in our study, with more than 80% of HIV-specialists who responded to our survey indicating that they were located in cities with more than 100,000 and 50% indicating that they were located in cities with more than 1 million inhabitants. Conversely, more than 50% of the non-HIV-specialists participating in our study reported that they were located in cities with fewer than 100,000 inhabitants. Any opportunity to increase the number of non-HIV-specialists who can give advice on PrEP and prescribe PrEP to patients at risk of acquiring HIV in conformity with the relevant guidelines should therefore be explored. The same can be said of the gap between the western and eastern German states more generally, where a decades-long tradition of large HIV-specialty practices and community-based counselling centers in the west contrasts with a lack of such facilities and institutions in the east.

As expected, our results suggest that HIV-specialists have greater knowledge and counseling competence related to PrEP, as well as more positive attitudes towards it, than do non-HIV-specialists. Unsurprisingly, a greater proportion of patients who had an indication for PrEP were proactively given advice on it by the HIV-specialists. This being said, attitudes towards PrEP and particularly knowledge of it were much more heterogeneous among our participating non-HIV-specialists than was the case among HIV-specialists, which suggests that at least some of the non-HIV-specialists in our sample might require little or no training on PrEP care. Indeed, the results of our multiple linear regression suggest that knowledge of PrEP was the only statistically significant predictor of the proportion of indicated patients who were proactively given advice and counseling on PrEP by participating physicians. It might therefore be wise for policymakers and other actors in the German health system to consider providing non-HIV-specialists who fit this description, particularly if they are in a rural location, with ways to demonstrate and certify their skills that are less onerous than those at present. At the same time, our data strongly suggest that there is indeed a need to provide training on PrEP to a very large percentage of non-HIV-specialists. On average, this group of respondents had less knowledge and poorer counseling skills with regard to PrEP care, as well as attitudes towards PrEP that were more negative than those reported by HIV-specialists. Non-HIV-specialists in our sample also reported providing pro-active counseling on PrEP to a much smaller proportion of individuals who had an indication for it than did HIV-specialists.

Even if non-HIV-specialists do actively refer patients to PrEP-certified physicians, this still requires them to be able to identify patients with an indication for PrEP and proactively discuss the topic. If the gap between rural and urban areas in Germany (and elsewhere) is to be narrowed in this regard, it will be essential to improve training to these physicians, but to do so in a way that takes better account of the local health infrastructure and geographical barriers, such as long distances to the nearest HIV specialty practices. Online training modules or telemedicine visits are just two options. Certainly, efforts in this direction would be welcomed by the participants in our sample, particularly by the non-HIV-specialists, about 62% of whom indicated that they wished to receive training or information materials on managing PrEP patients. Such training could be augmented by providing the participating physicians with information materials and decision aids for patients in patient-understandable language and in different languages. Indeed, in our survey, decision aids for patients were reported by participating physicians to be the materials they thought would increase the practicability of their PrEP counselling and prescriptions the most. Doing so would be a low-cost and potentially

efficient and effective way to augment the counseling skills of physicians who do not (yet) feel themselves to be competent enough to advice patients on taking PrEP.

There are some interesting similarities between the results of our survey and those of an earlier survey we conducted among counselors in community-based non-governmental STI/HIV counseling centers and local health offices [19]. In the latter, we also found differences in knowledge and attitude scores between the different organizational contexts, with the counseling centers having higher scores in both domains and a much larger proportion of LGBTI+ clientele compared to the local health offices–mirroring in some respects the gaps between HIV-specialists and non-HIV-specialists observed in the present study. Moreover, it is interesting that in the present study, as in our earlier survey, a substantial percentage of participants indicated that it would be helpful to have a clinical practice guideline that contained a clear presentation of indications, contraindications and necessary laboratory tests for PrEP. Given that a guideline on these subjects has, in fact, already been available since 2018, the substantial percentage of participants reporting a wish for such a guideline suggests that the dissemination and implementation of the guideline have been inadequate or that the guideline does not present the relevant information in a clear enough manner.

## Limitations

This study has a number of important limitations beyond its observational, cross-sectional design and the obvious caveats that this entails. First, the rate of response to the survey, at 5.53%, was very low. Such response rates are not uncommon in surveys of office-based health professionals, such as GPs or dermato-venereologists, in Europe [21], and knowing this we took extensive efforts to encourage participation in the survey by offering it in different formats and sending email reminders. Nevertheless, the low response rate means that our results are probably not representative of the broader populations of HIV-specialists and non-HIV-specialists in Germany and cannot be easily generalized to them. Along these lines, selection bias is a second potential limitation of this paper. Physicians with either profound or no knowledge of PrEP, and physicians with strongly positive or strongly negative attitudes towards it, may have been more passionate about the subject and therefore more likely to participate. While it is impossible to quantify this bias, it is reasonable to assume that those who were more ambivalent about PrEP were less likely to participate and should therefore be targeted more strongly in any future research of this nature. A third limitation of our study was our use of a self-developed questionnaire that, for pragmatic reasons, did not use validated constructs to measure knowledge and attitudes. There is ample evidence that there often exists a discrepancy between reported knowledge and skills and respondents' actual knowledge and skills [22]. A fourth important limitations is our grouping of MSM and transgender patients for pragmatic purposes, particularly related to the length of the study questionnaire. Differentiating between these two groups would have allowed us to obtain meaningful data on the barriers faced by transgender patients wishing to initiate PrEP, but would have gone beyond the scope of our study. Furthermore, we did not specifically include other populations at risk of acquiring HIV, such as intravenous drug users or sex workers, in our survey in order to increase the participation rate by keeping the questionnaire as short and feasible as possible.

A fifth limitation is our decision not to explore race- or migration-related barriers to PrEP initiation. While a lack of language-relevant materials was listed as a potential barrier and materials in various languages were thought to be helpful by physicians, migrant-specific or race-specific barriers, for example related to discrimination, were not examined. Studies from the US suggest that there are large discrepancies between Black, Indigenous Patients of Color (BIPOC) and white patients with regard to PrEP and antiretroviral uptake [23, 24]. Data on

this subject are sparse, but the discrepancies are likely to be considerable [25]. Given that a substantial proportion of new HIV cases in Germany is among migrants and it is unclear whether the infections have occurred abroad or within Germany [10], it will be crucial in future research to examine structural discriminatory practices that might hamper these individuals' access to appropriate PrEP care. Lastly, the sexual orientation of respondents was not examined in this questionnaire; however, it may play a role in counseling practices, as well as in the choice of whether to specialize in the care of patients living with HIV and of LGBTI + individuals more generally.

## Conclusions

The findings of this study on HIV-specialists' and non-HIV-specialists' knowledge of PrEP, their attitudes towards it, and their PrEP counseling practices in Germany point to opportunities to improve PrEP implementation in individuals at risk of acquiring HIV. The large gap between the two groups of physicians with regard to knowledge about and attitudes towards PrEP could be addressed, in part, by providing physicians with patient-centered information material. Online training modules or telemedicine visits may also represent more accessible training options, particularly in rural areas, where few HIV specialists are available. Furthermore, the existing guideline on PrEP should be re-evaluated in terms of its dissemination, implementation and ease of use.

## Supporting information

**S1 File. Survey questionnaire (Original German version).** Original German online version of the questionnaire used in the present survey.
(PDF)

**S2 File. Survey questionnaire (English translation).** English translation of the online version of the questionnaire used in the present survey. Please note that the present translation has been undertaken for the publication only).
(PDF)

**S3 File. Minimal underlying data set and codebook.** Age of respondents, postal code, state and qualitative data have been removed to ensure anonymity.
(XLSX)

## Author Contributions

**Conceptualization:** Mary Katherine Sammons, Matthew Gaskins, Frank Kutscha, Alexander Nast, Ricardo Niklas Werner.

**Formal analysis:** Mary Katherine Sammons, Matthew Gaskins, Ricardo Niklas Werner.

**Investigation:** Mary Katherine Sammons, Ricardo Niklas Werner.

**Methodology:** Mary Katherine Sammons, Matthew Gaskins, Frank Kutscha, Ricardo Niklas Werner.

**Project administration:** Mary Katherine Sammons, Ricardo Niklas Werner.

**Supervision:** Matthew Gaskins, Alexander Nast, Ricardo Niklas Werner.

**Writing – original draft:** Mary Katherine Sammons.

**Writing – review & editing:** Matthew Gaskins, Frank Kutscha, Alexander Nast, Ricardo Niklas Werner.

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
