## [Decision Letter · Decision Letter 0]

9 Apr 2021

PONE-D-21-02667

HIV Pre-exposure Prophylaxis (PrEP): Knowledge, Attitudes and Counseling Practices among Physicians in Germany – A cross-sectional survey

PLOS ONE

Dear Dr. Werner,

Thank you for submitting your manuscript to PLOS ONE. After careful consideration, we feel that it has merit but does not fully meet PLOS ONE’s publication criteria as it currently stands. Therefore, we invite you to submit a revised version of the manuscript that addresses the points raised during the review process.

Your manuscript has undergone the peer-review process and the reviewers have provided their comments/suggestions. Kindly address these points/concerns before we make a decision.

We look forward to receiving your revised manuscript.

Kind regards,

Kingston Rajiah

Academic Editor

PLOS ONE

Journal Requirements:

Please provide additional details regarding participant consent. In the ethics statement in the Methods and online submission information, please ensure that you have specified (1) whether consent was informed and (2) what type you obtained (for instance, written or verbal, and if verbal, how it was documented and witnessed). If your study included minors, state whether you obtained consent from parents or guardians. If the need for consent was waived by the ethics committee, please include this information.

If you are reporting a retrospective study of medical records or archived samples, please ensure that you have discussed whether all data were fully anonymized before you accessed them and/or whether the IRB or ethics committee waived the requirement for informed consent. If patients provided informed written consent to have data from their medical records used in research, please include this information."

Please include additional information regarding the survey or questionnaire used in the study and ensure that you have provided sufficient details that others could replicate the analyses. For instance, if you developed a questionnaire as part of this study and it is not under a copyright more restrictive than CC-BY, please include a copy, in both the original language and English, as Supporting Information.

Reviewers' comments:

Reviewer's Responses to Questions

**Comments to the Author**

1. Is the manuscript technically sound, and do the data support the conclusions?

Reviewer #1: Yes

Reviewer #2: Yes

2. Has the statistical analysis been performed appropriately and rigorously? 

Reviewer #1: Yes

Reviewer #2: Yes

3. Have the authors made all data underlying the findings in their manuscript fully available?

Reviewer #1: Yes

Reviewer #2: Yes

4. Is the manuscript presented in an intelligible fashion and written in standard English?

Reviewer #1: Yes

Reviewer #2: Yes

5. Review Comments to the Author

Reviewer #1: The results of this study suggest opportunities to improve PrEP implementation among persons at risk for HIV infection. Focused teaching, especially for non-HIV specialists, and the provision of patient-centered information materials could help improve care for at-risk groups. In particular, doing so could be the case in remote rural areas where few HIV specialists are an available resource.

The paper is well written and well presented. The topic is of course of interest, especially in Germany and Authors should be lauded for their effort. Future research projects explaining whether the infections among migrants occurred abroad or within Germany is very valuable and in my opinion this is not only medically but also politically necessary. Potential optimization options against new HIV infections will also be clearer in this process.

Reviewer #2: The article by Sammons and colleagues, in a cross-sectional study evaluated, the knowledge, attitudes and practices of HIV-specialist and non-specialist physicians on the topic of HIV PrEP. Using a self-developed questionnaire, they were able to acquire significant data and make meaningful assertions and deductions. Importantly, they were able to identify difficulties in finding a physician to prescribe PrEP as an important barrier for patients to initiate prophylaxis. The limitations of the study particularly the low participation rate was highlighted. The study was elegantly presented. The following points minor will however need to be addressed

Why were the risk groups in the questionnaire limited to include only MSM and transgender/LGBT+ individuals? Other significantly at-risk groups such intravenous drug abusers and commercial sex workers were not included. This may have also influenced the exclusion of other subspecialties of physicians to partake in the study.

It appears there was no question asking the HIV non-specialists if they will be interested in undergoing the needed training to specialize as an expert?

This is especially important since the authors found that knowledge of PrEP was the only statistically significant predictor of proactively given advice and counseling on PrEP.

The authors should comment on this.

6. PLOS authors have the option to publish the peer review history of their article (what does this mean?). If published, this will include your full peer review and any attached files.

Reviewer #1: **Yes: **Dr. med. Nuran Abdullayev

Reviewer #2: No

---

## [Author Response · Author response to Decision Letter 0]

13 Apr 2021

Response to the academic editor’s and reviewers’ comments 

Manuscript: ‘HIV Pre-exposure Prophylaxis (PrEP): Knowledge, Attitudes and Counseling Practices among Physicians in Germany – A cross-sectional survey’ (PONE-D-21-02667)

> We would like to thank the academic editor and each of the reviewers for their careful reviews of our manuscript and their very helpful suggestions for improving it. We feel that the manuscript has benefitted greatly as a result. Please find our replies to each of their points below, including details on how we have implemented their suggestions.

Academic editor comments

> We have reviewed PLOS ONE’s style requirements again and have now ensured that the manuscript and file names meet these in full. Specifically, we have shortened the abstract to meet the 300 word limit specified in the submission guidelines and formatted the headings in the requested font size.

> We have added three sentences in the “Study Design” section to clarify how participant consent was obtained (lines 123-126), as follows: “All participants were older than 18 years. Participants in the online survey provided their written informed consent by ticking the box next to a statement that they had read the study information and agreed to participate in the study. For participants who completed the paper version of the survey, we assumed consent if they returned their questionnaire by fax or mail.”

3. Please include additional information regarding the survey or questionnaire used in the study and ensure that you have provided sufficient details that others could replicate the analyses. For instance, if you developed a questionnaire as part of this study and it is not under a copyright more restrictive than CC-BY, please include a copy, in both the original language and English, as Supporting Information..

> Along with our revised documents, we have now uploaded our questionnaire as supporting information, both in the original German and in an English translation.

> We have reviewed the reference list and ensured that it is complete and correct.

Reviewer #1: 

The results of this study suggest opportunities to improve PrEP implementation among persons at risk for HIV infection. Focused teaching, especially for non-HIV specialists, and the provision of patient-centered information materials could help improve care for at-risk groups. In particular, doing so could be the case in remote rural areas where few HIV specialists are an available resource.

The paper is well written and well presented. The topic is of course of interest, especially in Germany and Authors should be lauded for their effort. Future research projects explaining whether the infections among migrants occurred abroad or within Germany is very valuable and in my opinion this is not only medically but also politically necessary. Potential optimization options against new HIV infections will also be clearer in this process.

> Thank you very much for your positive feedback.

Reviewer #2: 

The article by Sammons and colleagues, in a cross-sectional study evaluated, the knowledge, attitudes and practices of HIV-specialist and non-specialist physicians on the topic of HIV PrEP. Using a self-developed questionnaire, they were able to acquire significant data and make meaningful assertions and deductions. Importantly, they were able to identify difficulties in finding a physician to prescribe PrEP as an important barrier for patients to initiate prophylaxis. The limitations of the study particularly the low participation rate was highlighted. The study was elegantly presented. The following points minor will however need to be addressed

> Thank you for reading our paper so carefully and providing these helpful comments. 

Why were the risk groups in the questionnaire limited to include only MSM and transgender/LGBT+ individuals? Other significantly at-risk groups such intravenous drug abusers and commercial sex workers were not included. This may have also influenced the exclusion of other subspecialties of physicians to partake in the study.

> We agree with the reviewer that there are further populations at risk of acquiring HIV, apart from the group of MSM and transgender persons that we focused on in our survey. We did not include intravenous drug users or sex workers as specific risk groups in our survey for pragmatic reasons. To increase the participation rate, we aimed to keep the survey as short and feasible as possible. MSM and trans persons constitute the main group of individuals at risk of acquiring HIV in Germany. However, to point out this shortcoming of our questionnaire, we have included a sentence in the discussion section (lines 421 – 423), which reads as follows: “Furthermore, we have not explicitly included other populations at risk of acquiring HIV, such as intravenous drug users or sex workers, in our survey in order to increase the participation rate by keeping the questionnaire as short and feasible as possible.”

It appears there was no question asking the HIV non-specialists if they will be interested in undergoing the needed training to specialize as an expert?

This is especially important since the authors found that knowledge of PrEP was the only statistically significant predictor of proactively given advice and counseling on PrEP.

The authors should comment on this..

> Indeed, we did not include a question in our questionnaire to ask the participants whether they would be interested in undergoing the training currently required to be allowed to bill for PrEP services. However, we did ask both HIV specialists and non-HIV specialists whether educational material or training on various aspects of PrEP management and identifying PrEP candidates would be helpful for their practice. The findings are presented in the results section (lines 307 – 315). Because of the length of the manuscript, we had not presented the data separately for HIV specialists and non-specialists. However, we strongly agree with the reviewer that differentiating here between specialists and non-specialists is important, and have therefore added some sentences to specify this information (lines 310 – 313). We have also adapted a sentence in the discussion section accordingly (lines 376 – 377).

---

## [Editor Report · Decision Letter 1]

16 Apr 2021

HIV Pre-exposure Prophylaxis (PrEP): Knowledge, Attitudes and Counseling Practices among Physicians in Germany – A cross-sectional survey

PONE-D-21-02667R1

Dear Dr. Werner,

We’re pleased to inform you that your manuscript has been judged scientifically suitable for publication and will be formally accepted for publication once it meets all outstanding technical requirements.

Kind regards,

Kingston Rajiah

Academic Editor

PLOS ONE
---

## [Editor Report · Acceptance letter]

20 Apr 2021

PONE-D-21-02667R1 

HIV Pre-exposure Prophylaxis (PrEP): Knowledge, Attitudes and Counseling Practices among Physicians in Germany – A cross-sectional survey 

Dear Dr. Werner:

I'm pleased to inform you that your manuscript has been deemed suitable for publication in PLOS ONE. Congratulations! Your manuscript is now with our production department. 

Kind regards, 

on behalf of

Dr. Kingston Rajiah 

Academic Editor

PLOS ONE